# Coverage with Timely Administered Vaccination against Hepatitis B Virus and Its Influence on the Prevalence of HBV Infection in the Regions of Different Endemicity

**DOI:** 10.3390/vaccines9020082

**Published:** 2021-01-23

**Authors:** Karen K. Kyuregyan, Vera S. Kichatova, Olga V. Isaeva, Ilya A. Potemkin, Elena Yu. Malinnikova, Maria A. Lopatukhina, Anastasia A. Karlsen, Fedor A. Asadi Mobarhan, Eugeniy V. Mullin, Olga S. Slukinova, Margarita E. Ignateva, Snezhana S. Sleptsova, Elena E. Oglezneva, Elena V. Shibrik, Maria G. Isaguliants, Mikhail I. Mikhailov

**Affiliations:** 1Department of Viral Hepatitis, Russian Medical Academy of Continuous Professional Education, 125993 Moscow, Russia; vera_kichatova@mail.ru (V.S.K.); isaeva.06@mail.ru (O.V.I.); axi0ma@mail.ru (I.A.P.); malinacgb@mail.ru (E.Y.M.); a.carlsen@yandex.ru (A.A.K.); michmich2@yandex.ru (M.I.M.); 2Laboratory of Viral Hepatitis, Mechnikov Research Institute for Vaccines and Sera, 105064 Moscow, Russia; marialopatukhina@yandex.ru (M.A.L.); 1amfa@bk.ru (F.A.A.M.); caesusjuliar@yandex.ru (E.V.M.); chvtku.ssv@yandex.ru (O.S.S.); 3The Sakha Republic (Yakutia) Regional Department of Rospotrebnadzor, 677027 Yakutsk, Russia; yakutia@14.rospotrebnadzor.ru; 4Medical Institute, M.K. Ammosov North-Eastern Federal University, 677010 Yakutsk, Russia; sssleptsova@yandex.ru; 5Belgorod Regional Department of Rospotrebnadzor, 308023 Belgorod, Russia; orgotd@31.rospotrebnadzor.ru; 6Medical Faculty, Belgorod State National Research University, 308015 Belgorod, Russia; 7Department of Health and Social Protection of the Population of Belgorod Region, 308005 Belgorod, Russia; shibrik@belzdrav.ru; 8Department of Microbiology, Tumor and Cell Biology, Biomedicum, Karolinska Institute, 17165 Solna, Sweden; maria.issagouliantis@ki.se; 9Laboratory of Molecular Pathogenesis of Chronic viral Infections, NF Gamaleja Research Center of Epidemiology and Microbiology, 123098 Moscow, Russia; 10Research Department, Riga Stradins University, LV-1007 Riga, Latvia

**Keywords:** hepatitis B vaccine, birth dose coverage, HBV prevalence, hepatitis B epidemiology, public health

## Abstract

Universal hepatitis B vaccination of newborns was implemented in Russia starting from 1998. From 1998 to 2019, the incidence of acute hepatitis B reduced from 43.8 to 0.57 cases per 100,000 population. Here, we assessed the timely coverage of newborns with the birth dose (HepB-BD), second dose (HepB-2nd), and three vaccine doses (HepB3) in two remote regions of Russia with low (Belgorod Oblast) and high (Yakutia) levels of hepatitis B virus (HBV) endemicity. Vaccination data were obtained from the medical records of 1000 children in Yakutia and 2182 children in Belgorod Oblast. Sera of healthy volunteers from Belgorod Oblast (*n* = 1754) and Yakutia (*n* = 1072) across all age groups were tested for serological markers of HBV to assess the infection prevalence and herd immunity. Average HepB-BD coverage was 99.2% in Yakutia and 89.4% in Belgorod Oblast (*p* < 0.0001) and in both regions varied significantly, from 66% to 100%, between medical centers. The principal reason for the absence of HepB-BD was parent refusal, which accounted for 63.5% of cases of non-vaccination (83/123). While timely HepB-2nd coverage was only 55.4%–64.7%: HepB3 coverage by the age of one year exceeded 90% in both study regions. HBV surface antigen (HBsAg) prevalence in the 1998–2019 birth cohort was 0.2% (95% CI: 0.01–1.3%) in Belgorod Oblast and 3.2% (95% CI: 1.9–5.2%) in Yakutia. The proportion of persons testing negative for both antibodies to HBsAg (anti-HBs) and antibodies to HBV core antigen (anti-HBc) in the 1998–2019 birth cohort was 26.2% (125/481) in Belgorod Oblast and 32.3% (162/501) in Yakutia. We also assessed the knowledge of and attitude towards vaccination among 782 students and teachers of both medical and non-medical specialties from Belgorod State University. Only 60% of medical students knew that hepatitis B is a vaccine-preventable disease. Both medical and nonmedical students, 37.8% and 31.3%, respectively, expressed concerns about safety and actual necessity of vaccination. These data indicate the need to introduce a vaccine delivery audit system, improve medical education with respect to vaccination strategies and policies, and reinforce public knowledge on the benefits of vaccination.

## 1. Introduction

Hepatitis B caused by human hepatitis B virus (HBV) is a widespread, socially significant infection causing up to 900,000 deaths annually, primarily due to the underlying liver disease culminating in liver cirrhosis and hepatocellular carcinoma (HCC) [1]. Hepatitis B can be prevented by vaccination. Highly effective recombinant vaccines have been available since 1982 [2]. The most effective approach to control hepatitis B is universal newborn vaccination, with the first dose administered 24 h after birth. As of 2015, the hepatitis B neonatal vaccination program has been adopted in 185 countries [1]. In Russia, this program was implemented in 1998 and implied the vaccination of all newborns by standard immunization schedule (0, 1, and 6 months), except for children born to HBV infected mothers, who are vaccinated with four vaccine doses (0, 1, 2 and 12 months). For all newborns, regardless of mother’s HBV status, the first dose of vaccine is administered within 24 h after birth. In addition, about 62.5 million people, including adults under the age of 60, have been vaccinated under the National Priority Health Project launched in 2006. Within the framework of this project, free vaccination against hepatitis B was offered to all children, adolescents and adults under the age of 60. As a result of the project, by 2013 the coverage by three doses of hepatitis B vaccine reached 97% in children under 18, and 72% in adults aged 18–59 years [3]. This has led to a sharp decline in the incidence of acute hepatitis B, in other words, the number of new HBV infections. Acute hepatitis B incidence rates in the total population of Russia dropped from up-to 43 per 100,000 in the pre-vaccination period to the lowest indicators of 0.9 per 100,000 in 2016–2018 and 0.6 per 100,000 in 2019. In children under 18 years of age, the incidence of acute hepatitis B is even lower, 0.05–0.08 in 2017–2019 [4]. Despite being very low, these incidence rates speak of vaccination gaps resulting in continuous HBV circulation with possible mother-to-child transmission. Analysis of the effectiveness of the already implemented measures and recommendations of additional interventions are needed to achieve the World Health Organization (WHO) hepatitis elimination goals defined as 90% reduction in the incidence and 65% reduction in the mortality compared with the baseline of 2015 [5].

During the last 10 years, the coverage of hepatitis vaccination in children in Russia has been as high as 90%. However, this figure is based on the coverage with the complete vaccination during the first year of life. It does not account for the coverage with the most important first hepatitis B vaccine dose (birth dose; HepB-BD) [6]. The data on this key indicator for interventions under the WHO global health sector strategy on viral hepatitis [5] in Russia is limited.

The primary objective of our study was to determine the actual rates of coverage with timely HepB-BD and the complete vaccination course in two regions of Russia, one with a low level, and the other with a high level of HBV prevalence. The first, Belgorod Oblast, is localized in the European, and the other, Sakha Republic (Yakutia), in the Asian part of the country. Belgorod Oblast is typical of the European part of Russia in terms of economic and demographic characteristics; the reported annual rates of acute and chronic hepatitis B incidence are 20–40% lower than the average for the Russian Federation. In Yakutia, a region spanning across a large area of Central and Northern Siberia, hepatitis B presents a serious health care problem. In the pre-vaccination period, HBV prevalence in Yakutia was 3–4 times higher than the average for the Russian Federation [7]. In 2017–2019, the incidence of acute hepatitis B in Yakutia dropped to 0.2–0.4 per 100,000, however, the annual rates of chronic hepatitis B are still very high (24.9 per 100,000 in 2019) [4], pointing at existence of a large reservoir of the potential sources of HBV infection of the resident population.

Along with vaccination coverage in the studied regions, we assessed the prevalence of markers of current and past HBV infection among a conventionally healthy population in these regions in order to determine the epidemiological background and the impact of vaccination coverage on HBV prevalence. In the same cohorts, we analyzed herd immunity to HBV, both vaccine-induced, confirmed by the presence of antibodies to HBsAg (anti-HBs) only, and post-infection, indicated by reactivity for both anti-HBs and antibodies to HBV core antigen (anti-HBc), or for only anti-HBc. In addition, we conducted a survey of university students and teachers (including but not limited to medical faculty) in one of the study regions regarding their attitude to vaccination, to identify the factors that affect coverage with vaccination in practice.

## 2. Materials and Methods

### 2.1. Analysis of Hepatitis B Vaccination Coverage in Newborns

The study analyzed the coverage with timely administration of HepB-BD (given within 24 h after birth), timely administration of the second dose (HepB-2nd), given within 1–1.5 months after the timely birth dose, and the coverage with the complete three-dose vaccination (HepB3) within the first year of life, in children born in Yakutia and Belgorod Oblast, Russian Federation, during 2017–2018. The study design was approved by the Ethics Committee of the Mechnikov Research Institute for Vaccines and Sera in Moscow, Russia (Approval #1 dated 28 February 2018). Information about vaccination was obtained from the randomly selected individual vaccination history cards or from the individual copies of Form No.63 (a document that is generated for each child at his or her first visit to a pediatric polyclinic based on data transmitted from maternity hospitals). Written consent to participate in the study were signed by all parents of the vaccinated children.

Data were collected from three types of clinics, categorized based on their location: (1) in the regional capital; (2) in the district centers; and (3) in the rural areas, further referred to as city, town and rural clinics, respectively. In total, the study involved 10 clinics in Belgorod Oblast (4 city and 6 town clinics) and 16 centers in Yakutia (4 city, 7 town and 5 rural clinics) (Figure 1). Hepatitis B vaccination data were collected for 2182 newborns in Belgorod Oblast and 1000 newborns in Yakutia. The sample sizes represent 16.5% of 13,183 newborns in Belgorod Oblast, and 7.8% of 12,818 newborns in Yakutia (according to Russian Statistics Agency (Moscow, Russia) [8]. For all cases of missed HepB-BD, the reasons for non-vaccination, recovered from the medical records, were attributed to parents’ refusal or medical contraindications, and classified accordingly.

### 2.2. Serum Samples

The serum samples of 1754 healthy volunteers from Belgorod Oblast and 1072 healthy volunteers from Yakutia were collected in 2018–2019, i.e., 20 years post the on-start of the universal HBV vaccination of newborns in Russia. Both cohorts represent about 0.1% of the total population of the respective regions (approximately 967,000 persons in Yakutia and 1,550,000 in Belgorod Oblast; Bulletin of the Russian Statistics Agency [9]. Participants of the survey were selected randomly from persons undergoing routine medical examination, visitors to the vaccine room for routine vaccinations, patients visiting the clinic for reasons not related to infectious diseases. The population sample size was calculated for the known size of the general population of study regions based on the previously published data on HBV surface antigen (HBsAg) and anti-HBc prevalence rates in the Russian Federation [10] with chosen power (80%) and confidence level (95%) [11]. Serological survey was conducted according to the principles expressed in the Declaration of Helsinki. Design of the serological study was approved by the Ethics Committee of the Mechnikov Research Institute for Vaccines and Sera in Moscow, Russia (Approval #1 dated 2018-28-02). Written informed consent was obtained from all study participants. Serum samples were coded and stored in aliquots at −70 °C until testing. After thawing, sera were tested for the serological markers of HBV infection (Figure 1).

The subjects were males or females aged 0–95 years, conditionally healthy people with no symptoms of acute disease at enrollment (self-reported or parent-reported) and permanent residents of the study regions. Exclusion criteria were acute illnesses, a body temperature over 37.1 °C or any surgery, blood transfusion or treatment with blood products within three months before enrollment into the study. The study included eight age cohorts, from children under 1 year to seniors over 60 years of age (0–9, 10–14, 15–19, 20–29, 30–39, 40–49, 50–59, and ≥60 years). The mean population sample size in each age group was 219 individuals (90–382) in Belgorod Oblast and 153 (99–402) in Yakutia. The male/female ratio varied from 1:0.8 to 1:2, depending on the age cohort and study region. The mean age of participants was 44.2 ± 22.8 years in Belgorod Oblast, and 27.9 ± 21.3 years in Yakutia. The rural/urban population ratio was 1:6 both in Belgorod Oblast and in Yakutia.

### 2.3. HBV Testing

All serum samples from conditionally healthy individuals were tested for HBsAg, anti-HBc and anti-HBs using commercially available enzyme-linked immunosorbent assay (ELISA) kits (Vector-Best, Novosibirsk, Russia). The sensitivity of the HBsAg ELISA Kit, according to the manufacturer’s specifications, was 0.01 IU/mL. Samples were considered positive for anti-HBs if they contained antibodies in concentration of ≥10 mIU/mL. All samples positive for HBsAg were tested for HBeAg and antibodies to the hepatitis D virus (anti-HDV) using ELISA kits from the same manufacturer. All testing was performed according to the manufacturer’s instructions for the respective kits.

### 2.4. Survey of University Students and Teachers on Attitude to Vaccination

A total of 782 students and teachers of the Belgorod State University were interviewed on their knowledge of and their attitude towards vaccination using a specially designed questionnaire (Appendix A). Of the participants, 88.9% were students and 11.1%, university teachers. The male/female ratio among participants was 1:1.6, the average age was 22.5 ± 8.9 years (15–79 years). Among the participants, 147 students and 7 teachers were affiliated to the Medical Faculty. Other specialties/disciplines were represented by follows: Humanitarian (31 students and 8 teachers), Natural Science (75 students and 8 teachers), and Technical (360 students and 63 teachers). Participation in the survey was voluntary and anonymous. All survey participants were interviewed verbally; answers were entered into the questionnaire by the interviewer during the survey. The questionnaire contained 12 questions regarding the responders knowledge about (i) vaccination in general, the hepatitis B vaccine in particular; (ii) diseases that can be prevented by vaccination; (iii) vaccines; and (iv) confidence in vaccination. Data collected by the questionnaire were transferred into an Excel database and subjected to the statistical analysis.

### 2.5. Statistical Analysis

Data analysis was performed using graphpad.com. Statistical analysis included the calculation of a 95% confidence interval (95% CI) and assessment of the significance of differences in values between groups using Fisher’s exact test (significance threshold *p* < 0.05).

## 3. Results

### 3.1. Hepatitis B Vaccination Coverage in Newborns

The rates of coverage for timely HepB-BD, HepB-2nd and HepB3 per region are shown in Table 1. In Belgorod Oblast the average rate of HepB-BD coverage was 89.4% (894/1000). Analysis of the reasons of missed timely HepB-BD showed that in 69% of cases (73/106) it was due to the refusal of parents to vaccinate their newborns for personal reasons. Vaccination was not performed for medical reasons in 31% of cases (33/106). The most common reasons for the medical refusal were prematurity, neonatal jaundice, hemolytic disease of the newborn, perinatal CNS damage, and combination of these conditions with other pathologies. In 3 out of 33 cases, mothers declined to vaccinate also after one month, when medical reasons for non-vaccination subsided.

Cases with no HepB-BD were documented in all eight surveyed medical institutions in Belgorod Oblast. Of these, 48.1% of cases (51/106) were concentrated in a single clinic. In this particular clinic, the main reason for not receiving timely HepB-BD was the refusal of parents to vaccinate children (in 90.2%; 46/51 of the cases). This clinic also demonstrated the lowest rates of coverage with HepB-BD, HepB-2nd and HepB3 (Table 1, Center B6): 66.0%, 16.3%, and 80.7%, respectively. An investigation of the abnormally low rates of vaccination coverage in this clinic revealed that an employee who was directly involved in organizing the vaccination process disseminated negative information regarding vaccination to the mothers of the newborns.

In Yakutia, 99.2% (2165/2182) of the surveyed children received timely HepB-BD (Table 1). This coverage indicator was significantly higher than the HepB-BD rate in Belgorod Oblast (*p* < 0.0001). A total of 17 cases not receiving timely HepB-BD were documented in 6 out of 16 surveyed medical centers in Yakutia. Among these cases, ten (59%) were due to parent refusal, and seven, to medical reasons.

The average HepB3 coverage by one year of age was >90% in both study regions. This indicator was between 86% and 100% in all surveyed medical centers, except for the aforementioned clinic B6 (Table 1). In both regions, the lowest rates were observed for the timely coverage with HepB-2nd. This indicator was below 70% in 13 out 24 surveyed clinics (Table 1), indicating that the second dose and, therefore, the third dose of vaccine were often administered with a delay. However, it should be noted that all children who received HepB3 before age 1 year, received a second dose of vaccine, either on time or with a delay.

We have also observed differences in hepatitis B vaccine coverage rates depending on the location of the medical centers. In Belgorod Oblast, rates of coverage with timely HepB-BD, timely HepB-2nd and HepB3 in city clinics were significantly higher than in the town clinics (95.3% vs. 80.5%, *p* < 0.0001; 62.1% vs. 45.2%, *p* < 0.0001; 93.8% vs. 87.0%, *p* = 0.0003, respectively) (Table 2).

In Yakutia, timely HepB-BD coverage rates were similar and above 99% in all types of centers. Meanwhile, the proportion of children who received timely HepB-2nd varied from 50.4 to 76.9%, depending on the location of the medical institution. In city centers, this indicator was significantly lower than in town centers (70.1%, *p* < 0.0001) and in rural centers (76.9%, *p* = 0.0048). The proportion of children who received HepB3 upon reaching the age of 12 months was significantly higher in town centers (95.3%) than in city centers (89%, *p* ≤ 0.0001) and in rural centers (91.9 %, *p* = 0.0096).

### 3.2. Prevalence of Hepatitis B Markers in a Conditionally Healthy Population

The size of the study cohorts corresponded to 0.11% of the total population of each region, being 1,549,151 people in Belgorod Oblast and 971,996 people in Yakutia [8].

Average HBsAg prevalence rates were significantly higher in Yakutia than in the Belgorod Oblast (2.2% (24/1072) vs. 0.5% (9/1754), *p* < 0.01, Fisher exact test). Data on the prevalence of HBsAg, anti-HBc and anti-HBs in each age group are shown in Figure 1 and Appendix A. With the exception of people over 60 years of age, HBsAg prevalence was consistently higher in Yakutia across all age groups. The HBsAg positivity rate for subjects over 60 years in Yakutia was similar to the positivity rate in the corresponding age group in Belgorod Oblast (1.7% and 1.6%, respectively; Figure 2A). The highest HBsAg positivity rate was observed in Yakutia in children aged 10–14 years (4.0% (5/125)), however, the difference between this rate and positivity rates in other age groups in Yakutia was not statistically significant (*p* > 0.05, Fisher exact test). All HBsAg-positive serum samples were negative for both HBeAg and anti-HDV. Additionally, we calculated HBsAg prevalence in women of childbearing age (15–49 years), which was 1.1% (95% CI: 0.2–3.2%; 3/283) in Yakutia and 0.2% (1/536) in Belgorod Oblast. In Yakutia, this figure was significantly lower compared to HBsAg prevalence in children aged 0–14 years (3.5% (4/402), *p* < 0.05, Fisher exact test). In the Belgorod Oblast, HBsAg prevalence in children aged 0–14 years was 0.4% (1/238) and tended to be higher compared to women of childbearing age, but not significantly (*p* > 0.05, Fisher exact test).

The average prevalence of anti-HBc was significantly higher in Yakutia than in Belgorod Oblast (29.4% (315/1072) vs. 17.1% (300/1754), *p* < 0.0001, Fisher exact test). Anti-HBc positivity rates in each age group were significantly higher in Yakutia than in Belgorod Oblast, with the exception of children in the 0–9 and 10–14 age groups, where seropositivity rates were similar in Yakutia and in Belgorod Oblast (*p* > 0.05 for both age groups). Seropositivity rates increased with age in both regions, but the increase started earlier in Yakutia, with the 15–19 age group, while in Belgorod the increase started with the 40–49 age group (Figure 2B). It should be noted that according to the questionnaire data, none of the study participants reported a history of jaundice and/or liver disease.

Figure 2C shows the proportion of samples that were non-reactive for anti-HBc, but reactive for anti-HBs, i.e., indicating the proportion of participants who had only vaccine-induced anti-HBs antibodies and were not exposed to HBV. In children under 15 years of age, the proportion of the population with only vaccine-induced anti-HBs antibodies varied between 56–67% independently of the region (Belgorod Oblast and Yakutia) or age of the children. The proportion of population with only vaccine-induced anti-HBs antibodies among adolescents aged 15–19 and adults aged 20–29 years in Yakutia was significantly lower than in Belgorod Oblast (*p* = 0.0056, and *p* = 0.0002, respectively; Fisher exact test) (Figure 2C). In older age groups, the proportion of anti-HBs positive, anti-HBc negative samples declined in both regions in a similar way, but to a significantly lower level in subjects aged 60 years and older in Belgorod Oblast than in Yakutia (6.3% vs. 31.4%, *p* < 0.0001, Fisher exact test; Figure 2C).

We have also assessed the proportion of individuals susceptible to HBV in every age group as those non-reactive for both anti-HBc and anti-HBs (Figure 2D). It was similar in individuals under 30 years of age in both Yakutia and Belgorod Oblast, varying from 21.6% to 35.3% (Figure 2D). In the older age groups, the proportion of potentially susceptible to HBV individuals remained at the level of 15–16% in Yakutia, but significantly increased in the Belgorod Oblast (to 45–60%; *p* < 0.0001, Fisher exact test; Figure 2D).

### 3.3. Attitude to Vaccination among University Students and Teachers

A complete comparative analysis of the responses to interview questions related to knowledge of and attitude towards vaccination of university students and teachers of medical versus non-medical specialties is presented in Table 3. Only 45.5% (71/156) of medical, and 26.1% (164/626, *p* < 0.0001) of non-medical students and teachers mentioned HBV as an infection that can be prevented by vaccination. Of the respondents who did not mention HBV as a vaccine-preventable infection, only 60% (51/85) of medical and 51.1% of non-medical participants (236/462, *p* = 0.1560) responded positively to the question “Have you heard that HBV can be prevented by vaccination?”

The subset of questions regarding attitude to vaccination demonstrated similarly high levels of ignorance and hesitancy in the respondents with a medical as with a non-medical background. Only 37.8% (59/156) of medical students and teachers stated that they have no fears concerning vaccination (Table 3, Question #6), similar to the proportion of participants of non-medical specialties who gave the same response (*p* = 0.1273, Fisher exact test). At the same time, both categories expressed the same degree of confidence in the quality of vaccines (Table 3, Question #11; 72.4% and 63.6%, respectively, *p* = 0.0477, Fisher exact test). Importantly, high proportion of both medical and non-medical respondents expressed their trust to the medical personnel performing vaccinations (Table 3, Question #10; 84.6% and 78.7%, respectively, *p* = 0.1177, Fisher exact test). At the same time, a considerable proportion of medical students and teachers were tolerant to those who refuse to vaccinate themselves and their children; among non-medical respondents, this proportion was even higher (Table 3, Question #9; 49.7% versus 70.4%, respectively; *p* < 0.0001, Fisher exact test). A lack of information on vaccines was mentioned by a majority of participants in both groups; only 7.9% (62/782) of respondents were satisfied with the information on the vaccines and vaccination available on the public and/or governmental domains.

## 4. Discussion

Vaccination of newborns against hepatitis B is the main tool for controlling this infection and is a cornerstone of the WHO global health sector strategy on viral hepatitis [5]. In 2015, global coverage of timely HepB-BD was 39%. To achieve the WHO elimination strategy targets, it should be increased to 50% in 2020, and 90% in 2030. Coverage with a complete vaccination with three doses (HepB3) is another key indicator of the successful immunization program, and to control HBV spread it must reach 90% by 2020 [1]. Normally, hepatitis B vaccination program does not involve monitoring of the antibody response, since HBV vaccine is highly immunogenic and does not require additional immunizations, at least in the immunologically competent individuals who have received HepB3 [12]. It is assumed that seroprotection rates following the hepatitis B vaccination of infants are equal to or only marginally lower than HepB3 coverage, as the complete course of immunization induces protective antibody concentrations in >95% of healthy infants, children and young adults [13,14]. However, data from the real-world practice indicate that the actual seroprotection rates are much lower. In our study, the proportion of individuals who did not have vaccine-induced or post-infection immunity among the vaccinated generation (birth cohort 1998–2019; where officially reported HepB3 coverage is >95%) was more than 25% [3]. Also, the proportion of children under 10 years of age who have the vaccine-induced immunity (are anti-HBc-negative and have not contracted the virus) was below 60%. This is unexpectedly low, even considering that anti-HBs may wane with time. In China, the anti-HBs positivity rate among children who did not receive any booster immunization was 60.9% on average and declined with age from to 75.2% in children born in 2002–2003 to 44.3% in those born in 2012–2013 [15]. In a study by Le et al., vaccine-induced immunity to HBV among US-born children varied from 60.7% to 65.2% in children aged 2 to 5 years and from 46.5% to 64.6% in children aged 6 to 10 years, depending on the birth cohort. Interestingly, the rates of vaccine-associated immunity in this study significantly decreased in the 1994–2003 birth cohort on contrary to those in the children vaccinated earlier (1988–1993 birth cohort) [16].

Detection rates of anti-HBc antibodies, indicating exposure to HBV, in the 1998–2019 birth cohort are alarmingly high, varying from 8% to 18%, depending on the geographical region and age group. Moreover, the prevalence of HBsAg in Yakutia is not significantly lower for those born between 1998 and 2019 when compared to older age groups, suggesting the limited impact of vaccination on HBV prevalence in the region. Such anti-HBc positivity rates in the vaccinated generation, together with relatively low HBsAg prevalence and the absence of a self-reported history of jaundice or liver disease, suggest that a significant proportion had a self-limited subclinical infection in vaccinated individuals, i.e., an ongoing circulation of HBV in the vaccinated generation born between 1998 and 2019. The plausible explanation of our data is vertical HBV transmission from the infected mothers. In Russia, testing of all pregnant women for HBsAg is mandatory. All HBsAg positive pregnant women are further tested for HBeAg and HBV viral load to assess the risk of vertical transmission. Unfortunately, records on HBV status of mothers of participants of this serosurvey at the time of delivery were not available. WHO recommends tenofovir prophylaxis of mother-to child transmission in pregnant women with HBV viral load above 200,000 IU/mL [17], but this effective measure is not yet applied in Russia. Neither is HBIG administration to children born to HBV infected mothers. The only intervention for children born to HBV infected mothers in Russia is four doses immunization at months 0, 1, 2 and 12 compared to standard vaccination schedule for children born to HBsAg negative mothers (0, 1 and 6 months) [18]. In this context, vertical HBV transmission cannot be excluded. However, the vertical transmission alone could not account for high HBsAg prevalence in the birth cohort born after the start of universal vaccination program in Yakutia. HBsAg positivity rates in adult population, including those observed in women of childbearing age, were lower compared to children aged 0–14 years. Thus, other transmission routes and risk factors may contribute to ongoing HBV circulation in children and adolescents, which requires further study.

Earlier, we showed that vaccination against hepatitis B in several regions of Russia, including Yakutia, led to a significant decrease in the incidence and prevalence of HBV in the general population and the number of expected unfavorable outcomes of the infection [10]. Unfortunately, today, ten years after this study, we see no further pronounced decline in HBsAg detection in the general population of Yakutia (although the sample size in our study might be not large enough to reliably characterize the exact prevalence of HBsAg or anti-HBc antibodies in every age cohort in the region). Furthermore, we observe relatively low anti-HBs detection rates in the vaccinated-at-birth cohort. Such low levels cannot be attributed to the low sensitivity of the test used, as the detection limit for our anti-HBs test was 1 mIU/mL, according to the manufacturer’s manual. Nor can such low seroprotection rates be due to waning of the vaccine-induced immunity, since the rates of anti-HBs seropositivity in children aged 10–14 were found to be similar or even slightly higher than in children aged 0–9 years.

It is also highly unlikely that insufficient immune response to HBV vaccination was due to the poor vaccine quality. According to the information provided by the agency responsible for the quality of products in the Russian Federation (Rospotrebnadzor) = HBV vaccination is performed with as many as six hepatitis B vaccines of the renown international manufacturers and five national vaccines, all licensed for use in the Russian Federation [19]. All vaccines are wildly used and recognized as interchangeable, with trials demonstrating similar immunogenicity [20]. Furthermore, each series of home-produced vaccines undergo the quality control, which includes, but is not limited to, the determination of sedimentation stability, pH, sterility, absence of bacterial endotoxins, pyrogenicity, toxicity, concentration of HBsAg in vaccine, immunogenicity tested in vivo, and completeness of antigen sorption [21].

One of the reasons for reduced vaccine-induced immunity to HBV could be partial inactivation of the vaccine preparations due to faults in the vaccine cold chain. Freezing a vaccine results in the loss of its immunogenicity [22,23]. Thus, vaccination in regions with harsh winters may be less effective during the winter due to improper transportation and storage conditions, as has been described for HBV vaccination performed in Mongolia [24].

The other highly plausible explanation for a bias between relatively low anti-HBs (and high anti-HBc antibody) and the officially reported HepB3 coverage could be vaccination gaps (uncompleted or not timely vaccinations), which facilitate mother-to-child or horizontal transmissions during the first year of life. The vaccination coverage data obtained in our study from vaccination records stored at clinics appear to be representative, since home birth is not a common practice in Russia, while the vast majority of births take place in maternity hospitals, except for indigenous nomadic population in some regions of the country [25]. The majority of children are not missed for the assignment to clinics and have vaccination records (Form 63), as the latter is a mandatory document for further admission to kindergarten and school. Thus, we assume the risk of the overestimation of the vaccination coverage to be minimal in this study. Our data from the real-world practice demonstrate that in some perinatal centers the coverage rates could be significantly lower than the average rates for the region, even for the region with satisfactory indicators of the average coverage. Even if HepB3 coverage reaches the WHO target indicator in infants by one year of age, rates of coverage with HepB-BD and timely administration of HepB-2nd can vary greatly between clinics and may be below the expected 90% coverage rate. Timely HepB-2nd coverage was only 55.4%–64.7%, suggesting that a delay in the vaccination schedule could be a widespread practice. In one particular clinic in this study, timely HepB-BD coverage was only 66%. We came across only one clinic with grossly malfunctioning vaccination practices, but there could be more. Timeliness of HepB-2nd was shown to be critical for protection of infants against horizontal HBV transmission [26]. Late provision of 2nd and 3rd vaccine doses puts children at a risk of HBV infection before receiving the complete vaccination course, specifically in the regions of high endemicity, where they may acquire HBV from the infected members of the household. Earlier studies revealed strong associations of HBV transmission among children with sharing of the personal objects (sharp objects, dishes, cutlery, glasses, face towels, and toothbrush) with HBV-infected family members [27], specifically mothers [28]. Importantly, continued HBV circulation creates favorable conditions for further spread of the virus to non-vaccinated individuals, and can potentially result in the spread in the 1998–2019 birth cohort of HDV, especially in Yakutia, where this infection is highly prevalent [29]. Altogether, published data and our findings stress the importance of strict adherence to vaccination schedule and the necessity of full HBV vaccination with three doses, to prevent this scenario.

Investigation of the reasons for the low rates of vaccination coverage in a particular clinic, demonstrated that a doctor directly involved in the vaccination process negatively impacted mothers’ decisions to vaccinate their newborns. It has been previously shown that health care providers have strong influence on the decision of parents to vaccinate their children, and that a positive influence results in significantly higher estimated coverage rates [30]. Our data further supports the significance of health workers’ influence on the frequency of parental refusal to vaccinate, and the necessity to train personnel organizing and performing HBV vaccination.

The average proportion of parental refusal in our study was as high as 67.5% among all cases of missed timely HepB-BD. Data from all over the world suggest that the main reasons for low HepB-BD coverage are parental concerns on vaccination safety and a lack of information [31,32,33]. To find out if this is true in the settings of this study, we conducted a survey of the university students and teachers, from both medical and non-medical specialties, aimed to characterize their attitudes to vaccines and vaccination, including vaccination against hepatitis B. Our survey revealed that alarmingly low proportion of medical students and teachers (only 60%) were aware that hepatitis B can be prevented by vaccination. Furthermore, the proportion of medical students and teachers concerned about vaccination safety amounted to 63%, being disturbingly similar to the proportion of non-professionals with the same opinion. These figures manifest strong vaccine hesitancy prevailing in different layers of the Russian population. Such hesitancy is a serious concern in the current epidemiological situation characterized by free circulation of HBV in the general population, as it endangers the implementation of hepatitis B elimination program. Health professional education programs need to be revised to emphasize the importance of vaccination, and hold healthcare providers accountable for participation in anti-vaccination campaigns and disseminating vaccination misinformation. Moreover, special vaccine audit programs aimed at random monitoring of vaccination coverage and determination of vaccine-induced herd immunity might be beneficial to estimate the true protection rates.

## 5. Conclusions

The most important result of our serosurvey is that the proportion of the protected individuals in the vaccinated generation in a region highly endemic for HBV is below expectations and insufficient to stop HBV circulation. Our data demonstrate the presence of gaps in the coverage of HepB-BD and timely administration of HepB-2nd in real-world practice despite high HEPB3 coverage in children by one year of age. Such gaps could be due to faults in the vaccine delivery cold chain and/or inadequate performance of the medical personal responsible for vaccination, but mainly are associated with the parental refusal to vaccinate their children. The latter, however, may result from the hesitancy towards vaccination of the health care providers. Our observations are supported by the lack of knowledge about hepatitis B vaccination and fears of vaccination demonstrated in the representative interviews done among medical students and teachers. Such settings lead to a reduced rate of HBV vaccination coverage resulting in an ongoing circulation of HBV among the vaccinated-at-birth cohort, both in the endemic and non-endemic regions of the Russian Federation. This unfavorable situation indicates the need to introduce a vaccine audit system aimed at monitoring hepatitis B vaccination coverage and vaccine-induced immunity, as well as strongly improve medical education in the field of vaccines and vaccinations.

## Figures and Tables

**Figure 1 vaccines-09-00082-f001:**
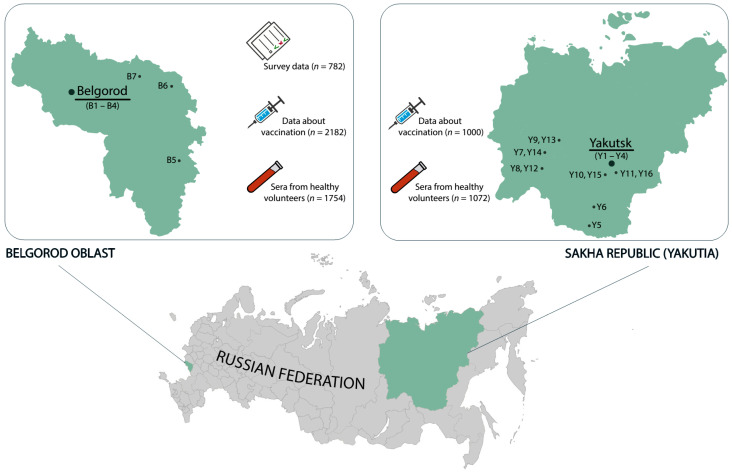
Study regions (in green) with indicated centers in each region where data on vaccination coverage were collected (B1 to B10–centers in Belgorod Oblast, Y1 to Y16–centers in Yakutia). The capital of each region is underlined. The number of participants in each step of the study (newborn vaccination coverage, serosurvey, survey of university students and teachers) are indicated in the callouts.

**Figure 2 vaccines-09-00082-f002:**
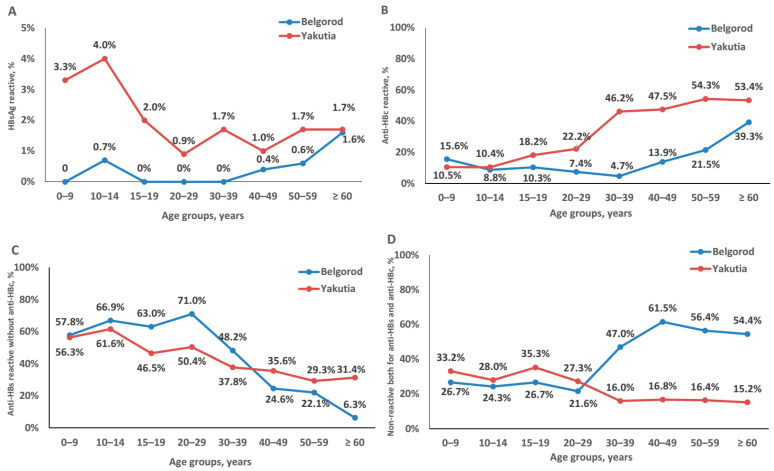
Proportions of serum samples reactive for hepatitis B markers in each age cohort of conditionally healthy individuals in the Belgorod Oblast and Yakutia: reactive for HBV surface antigen (HBsAg) (**A**); reactive for antibodies to HBV core antigen (anti-HBc) (**B**); reactive for antibodies to HBsAg (anti-HBs), but non-reactive for anti-HBc (**C**); non-reactive for either anti-HBs, or anti-HBc (**D**).

**Table 1 vaccines-09-00082-t001:** Rates of coverage with timely HepB-BD, timely HepB-2nd and HepB3.

Region	Clinic ID	Total Number of Children	Children Who Received HepB-BD		Children Who Received HepB-2nd at 4–6 Weeks		Children Who Received HepB3 before Age 1 Year	
			*N*	%	*N*	%	*N*	%
Belgorod Oblast	B1	100	94	94.0	51	51.0	99	99.0
	B2	100	98	98.0	90	90.0	98	98.0
	B3	100	93	93.0	67	67.0	93	93.0
	B4	300	287	95.7	165	55.0	273	91.0
	B5	100	98	98.0	54	54.0	98	98.0
	B6 *	150	99	66.0	25	16.3	121	80.7
	B7	50	42	86.0	35	70.0	43	86.0
	B8	100	83	83.0	67	67.0	86	86.0
	Total	1000	894	89.4	554	55.4	911	91.1
Yakutia	Y1	140	140	100	62	44.3	122	87.1
	Y2	208	205	98.6	83	39.9	187	89.9
	Y3	100	99	99.0	46	46.0	89	89.0
	Y4	363	359	98.9	218	60.1	324	89.3
	Y5	100	93	93.0	46	46.0	95	95.0
	Y6	100	100	100	54	54.0	97	97.0
	Y7	100	100	100	72	72.0	96	96.0
	Y8	100	100	100	86	86.0	96	96.0
	Y9	104	104	100	79	76.0	100	96.2
	Y10	148	148	100	108	73.0	133	89.9
	Y11	100	100	100	82	82.0	100	100
	Y12	100	100	100	80	80.0	89	89.0
	Y13	106	106	100	79	74.5	100	94.3
	Y14	100	99	99.0	63	63.0	93	93.0
	Y15	213	212	99.5	159	74.6	189	88.7
	Y16	100	100	100	95	95.0	98	98.0
	Total	2182	2165	99.2	1412	64.7	2008	92.0

* The center with the lowest vaccination coverage rates is indicated in red.

**Table 2 vaccines-09-00082-t002:** Rates of coverage with timely HepB-BD, timely HepB-2nd and HepB3 by location of the medical center.

Region	Type of Medical Center	Total Number of Children	Children Who Received HepB-BD		Children Who Received HepB-2nd at 4–6 Weeks		Children Who Received HepB3 before Age 1 Year	
			*N*	%	*N*	%	*N*	%
Belgorod Oblast	City	600	572	95.3	373	62.1	563	93.8
	Town	400	322	80.5	181	45.2	348	87.0
Yakutia	City	811	803	99.0	409	50.4	722	89.0
	Town	752	745	99.1	527	70.1	717	95.3
	Rural	619	617	99.7	476	76.9	569	91.9

**Table 3 vaccines-09-00082-t003:** Knowledge of and attitude towards vaccination expressed by university students and teachers of medical and non-medical specialties.

Questions	Medical Specialties		Non-Medical Specialties	
	*n*/*N* *	%	*n*/*N* *	%
1. Named vaccination among possible ways to prevent dangerous infections	126/156	80.8%	446/626	71.2%
2. Do you know what vaccination is? Answered “Yes”	150/156	96.2%	468/626	74.8%
3. Named hepatitis B as one of the infections prevented by vaccination (respondents were not prompted with a list of possible answers)	71/156	45.5%	164/626	26.1%
4. Knew that hepatitis B can be prevented by vaccination (targeted refinement to those who did not mention HBV in response to Question #3). Answered “Yes”	51/85	60.0%	236/462	51.1%
5. Heard about the National Vaccination Schedule. Answered ”Yes”	87/156	55.8%	229/626	36.6%
6. Vaccination fears (multiple answer choices. a list of possible answers was read out)				
a. Have no fears about vaccination	59/156	37.8%	196/626	31.3%
b. Children receive too many vaccines in the first two years of life	12/156	7.7%	62/626	9.9%
c. Vaccines can cause side effects or complications	84/156	53.8%	370/626	59.1%
d. Vaccines can weaken the immune system of children and adults	42/156	26.9%	235/626	37.5%
e. Vaccines can cause disease	31/156	19.9%	155/626	24.8%
f. The ingredients in vaccines are unsafe	12/156	7.7%	83/626	13.3%
g. The long-term effects of vaccines on humans have not been studied	10/156	6.4%	72/626	11.5%
h. Vaccination is against my religious beliefs	1/156	0.6%	7/626	1.1%
j. The quality of vaccines is low	4/156	2.6%	46/626	7.3%
k. Other	5/156	3.2%	21/626	3.4%
7. Have you or your relatives/acquaintances encountered illness/medical complications in children caused by vaccination? Answered “Yes”	61/156	39.1%	224/626	35.8%
8. Have you or your relatives/acquaintances encountered serious illness that could have been prevented by vaccination? Answered “Yes”	21/156	13.5%	94/626	15.0%
9. There are people/groups of people who refuse to vaccinate themselves and their children. How do you feel about their decision?				
a. I disagree. They risk the health of their children and contribute to the spread of disease	78/155	50.3%	189/617	30.6%
b. I do not care. They are entitled to their opinion	65/155	41.9%	354/617	57.4%
c. I agree with them	10/155	6.5%	62/617	10.0%
d. I haven’t heard anything about this	2/155	1.3%	6/617	1.0%
e. Other	0	0.0%	6/617	1.0%
10. Do you trust healthcare providers who vaccinate? (Choose one answer from the list)				
a. I completely trust them	53/156	34.0%	180/623	28.9%
b. I mostly trust them	79/156	50.6%	310/623	49.8%
c. No opinion	9/156	5.8%	37/623	5.9%
d. I mostly do not trust them	12/156	7.7%	81/623	13.0%
e. I do not trust them at all	3/156	1.9%	15/623	2.4%
11. Do you trust the quality of vaccines? (Choose one answer from the list)				
a. I completely trust the quality of vaccines	39/156	25.0%	129/619	20.8%
b. I mostly trust the quality of vaccines	74/156	47.4%	265/619	42.8%
c. No opinion	24/156	15.4%	96/619	15.5%
d. I mostly do not trust the quality of vaccines	14/156	9.0%	106/619	17.1%
e. I do not trust the quality of vaccines at all	5/156	3.2%	23/619	3.7%
12. What information about vaccinations would you like to receive? (Choose any answers from the list)				
a. Information about the usefulness and benefits of vaccination	97/156	62.2%	421/626	67.3%
b. Information about additional vaccines	81/156	51.9%	268/626	42.8%
c. Information about vaccine composition	89/156	57.1%	347/626	55.4%
d. Information about contraindications to vaccination	110/156	70.5%	483/626	77.2%
e. Information about complications after vaccination	107/156	68.6%	488/626	78.0%
f. Information about the manufacturers of vaccines	51/156	32.7%	269/626	43.0%
g. Other	1/156	0.6%	9/626	1.4%
h. I have enough information	17/156	10.9%	45/626	7.2%

* *n*—number of respondents who answered the question; *N*—number of respondents who were asked the question.

## Data Availability

The data presented in this study are available in this article.

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
