# Peer review of "Coverage with Timely Administered Vaccination against Hepatitis B Virus and Its Influence on the Prevalence of HBV Infection in the Regions of Different Endemicity"

_vaccines, 2021, doi:10.3390/vaccines9020082_

Round 1

Reviewer 1 Report

This is a very good serosurvey report on hepatitis B vaccination in Russia that focused on two approved vaccine strategies. The study outcomes are very informative for clinicians as well as health authorities. My major concern is the poor language control (few examples of suggested changes are highlighted in the attached file). Authors are advised to get the manuscript reviewed by a native speaker of English or professional editing service. The manuscript should be accepted upon proper revision.

Author Response

We are very grateful to Reviewer for comments and thorough analysis of our paper.

Comment 1

My major concern is the poor language control (few examples of suggested changes are highlighted in the attached file). Authors are advised to get the manuscript reviewed by a native speaker of English or professional editing service.

Response 1

Following Reviewer’s suggestion, we have modified the title of the manuscript as follows "Coverage with timely administered vaccination against hepatitis B virus and its influence on the prevalence of HBV infection in the regions of different endemicity". Furthermore, we subjected the revised version of the manuscript to additional revision by a native English-speaking researcher proficient in the field of vaccines and vaccination. Most of the respective changes can be seen as done by "Admin" in revised manuscript. Minor language changes were accepted and are not seen in tracking. We hope that Reviewer will find this language revision satisfactory, and the manuscript acceptable for publication.

Reviewer 2 Report

line 91: pointing at a huge depo of the potential 

replace with: pointing to a large reservoir of the potential

line 115:

Hepatitis B vaccination data were collected for 2,182 115 newborns in Belgorod Oblast and 1,000 newborns in Yakutia. 

It is important to define how were these samples selected, what percentage of the whole population they represent and why they are representative 

line 126:

of 1,754 healthy volunteers from Belgorod Oblast and 1,072 healthy 126 volunteers from Yakutia, same selection criteria to be applied as above

line 205, Table 2:

the coverage at week 4-6 weeks is significantly lower that the HepB3 figures. Although briefly mentioned in the discussion, there should be a clear indication from the records that HepB2 was administered, otherwise the seriousness of record keeping is in jeopardy, severely impacting the overall value of this article.

line 226, figure 2:

is the label of the ordinate in figure 2A correct?

line 328: the insufficient immune response is correctly addressed by mentioning the potential cold chain problems, gaps are less likely to be a possible explanation. However, no mention is made of the quality of the vaccine. It should be carefully mentioned to foster future improvements in quality controls.

Author Response

We are grateful to the Reviewer for the comments and thorough analysis of the manuscript.

Comment 1

line 91: pointing at a huge depo of the potential

replace with: pointing to a large reservoir of the potential

Response 1

We change the sentence accordingly (page 3, line 99).

Comment 2

line 115:

Hepatitis B vaccination data were collected for 2,182 newborns in Belgorod Oblast and 1,000 newborns in Yakutia.

It is important to define how were these samples selected, what percentage of the whole population they represent and why they are representative

Response 2

Information about vaccination was obtained from the randomly selected individual vaccination records. We did not calculate sample size for this particular part of the study. The proportion of newborns included in our study in two regions represents 16.5% and 7.8% of children born in 2019 in Belgorod Oblast and Yakutia, respectively (the total number of newborns in 2019 was 13,183 in Belgorod Oblast and 12,818 in Yakutia according to Russian Statistics Agency (https://rosstat.gov.ru/folder/12781). We believe that such proportion is representative for the whole population of newborns in the study regions.

We added this information to Materials and Methods section of the revised manuscript (page 3, lines 126-128).

Comment 3

line 126:

of 1,754 healthy volunteers from Belgorod Oblast and 1,072 healthy 126 volunteers from Yakutia, same selection criteria to be applied as above

Response 3

All participants of the survey were selected randomly. Both regional cohorts represent about 0,11% of total population of the respective regions, which is about 965,000 persons in Yakutia and 1,550,000 in Belgorod Oblast according to the Bulletin of the Russian Statistics Agency (http://www.gks.ru/free_doc/new_site/population/demo/Popul2019.xls).

The population sample size was calculated for the known size of the general population of study regions based on the previously published data on HBsAg and anti-HBc prevalence rates in the Russian Federation [Klushkina, V.V.; Kyuregyan, K.K.; Kozhanova, T.V.; Popova, O.E.; Dubrovina, P.G.; Isaeva, O.V.; Gordeychuk, I.V.; Mikhailov, M.I. Impact of universal hepatitis B vaccination on prevalence, infection-associated morbidity and mortality, and circulation of immune escape variants in Russia. PLoS One 2016, 11(6), e0157161. https://doi.org/10.1371/journal.pone.0157161.] with chosen power (80%) and confidence level (95%) [Hajian-Tilaki K. Sample size estimation in epidemiologic studies. Caspian J Intern Med. 2011;2(4):289–298.]. We added this information to Materials and Methods section of the revised manuscript (page 4, lines 140-147).

Comment 4

line 205, Table 2:

the coverage at week 4-6 weeks is significantly lower that the HepB3 figures. Although briefly mentioned in the discussion, there should be a clear indication from the records that HepB2 was administered, otherwise the seriousness of record keeping is in jeopardy, severely impacting the overall value of this article.

Response 4

Indeed, all those who had HepB3 by the end of the first year of life also received HepB2, according to medical records. Table 2 shows the proportion of timely HepB2nd. It means that those who had HepB3 (complete vaccination course of three doses) but did not have timely HepB2nd, have received the second dose of vaccine, but later during the first year of life. To clarify this, we added the indication that second dose was administered to all who received HepB3 (page 6, lines 222-223).

Comment 5

line 226, figure 2:

is the label of the ordinate in figure 2A correct?

Response 5

Thank you for noticing the typo, we corrected the label of Y-axis in figure 2A

Comment 6

line 328: the insufficient immune response is correctly addressed by mentioning the potential cold chain problems, gaps are less likely to be a possible explanation. However, no mention is made of the quality of the vaccine. It should be carefully mentioned to foster future improvements in quality controls.

Response 6

The poor vaccine quality does not seem to be a significant reason for insufficient immune response to vaccine, as according to Rospotrebnadzor, as many as six vaccines from international manufacturers and five home-produced vaccines against hepatitis B are licensed in Russia and their trials demonstrated similar immunogenicity [http://www.28.rospotrebnadzor.ru/activity/?p=1145&show_year=2008]. All these vaccines are wildly used and are recognized as interchangeable [Khotova T.Yu., Snegireva I.I., Darmostukova M.A., Zatolochina K.E., Ozeretskovskiy N.A., Shalunova N.V., Romanov B.K. The interchangeability of vaccines against hepatitis B for immunization of adults. Russian medical journal. 2016; 22 (2):85-89. doi 10.18821/0869-2106-2016-22-2-85-90. (in Russian)]. Moreover, each series of home-produced vaccines undergo the quality control, which includes, but not limited to, the determination of sedimentation stability, рН, sterility, absence of bacterial endotoxins, pyrogenicity, toxicity, concentration of HBsAg in vaccine, immunogenicity tested in vivo, and completeness of antigen sorption [Pharmacopoeia article "Hepatitis B vaccine recombinant. FS.3.3.1.0026.15, State Pharmacopoeia of the Russian Federation. XIII edition. Volume III"].

We added the respective paragraph to Discussion section of the revised manuscript (pages 11 and 12, lines 365-375).

Reviewer 3 Report

Dear Authors

I found your manuscript very interesting, mainly because the results and conclusions of this study could be support changes in Russian oficial policy regarding HepB vaccination.

I have some questions and comments

For this Birth cohort testing positive for HBsAg or anti HBc, do you have data about HBIG administration to newborns, or about their mothers  HBV profile(HBsAg, HBeAg, HBV DNA) or peripartum antiviral prophylaxis if they are HBsAg positive?

These Information are very important because the successful immunization programme  to eliminate the perinatal  mother to infant  HBV transmission or the orizontal transmission to the child in the first year of age depends equally on mothers testing and management and infant management.

Data from epidemiological studies and modelling indicate that high coverage of three- or four-dose infant vaccination, including timely birth dose, would not be sufficient to reach the incidence elimination goals by 2030 (HBsAg prevalence of <0.1% in children five years of age) Antiviral prophylaxis of pregnant women with high viral load may need to be added .

Countries that have not yet reached the 2020 goal of 1% HBsAg prevalence among children aged 5 years through vaccination would need to focus their efforts on increasing their vaccination coverage, including timely birth dose. All eligible pregnant and breastfeeding women living with HBV infection can safely use tenofovir

( WHO July 2020)

I will suggest to be accepted this manuscript  for publication with minor revision

Author Response

We are very grateful to Reviewer for comments and thorough analysis of our paper.

Comment 1

For this Birth cohort testing positive for HBsAg or anti HBc, do you have data about HBIG administration to newborns, or about their mothers HBV profile (HBsAg, HBeAg, HBV DNA) or peripartum antiviral prophylaxis if they are HBsAg positive?

These Information are very important because the successful immunization programme  to eliminate the perinatal  mother to infant  HBV transmission or the horizontal transmission to the child in the first year of age depends equally on mothers testing and management and infant management.

Data from epidemiological studies and modelling indicate that high coverage of three- or four-dose infant vaccination, including timely birth dose, would not be sufficient to reach the incidence elimination goals by 2030 (HBsAg prevalence of <0.1% in children five years of age) Antiviral prophylaxis of pregnant women with high viral load may need to be added.

Countries that have not yet reached the 2020 goal of 1% HBsAg prevalence among children aged 5 years through vaccination would need to focus their efforts on increasing their vaccination coverage, including timely birth dose. All eligible pregnant and breastfeeding women living with HBV infection can safely use tenofovir (WHO July 2020)

Response 1

Unfortunately, we do not have any data on HBV status of mothers of participants of this serosurvey. Thus, the vertical transmission could not be ruled out.  In Russia, the testing of all pregnant women for HBsAg is mandatory. All HBsAg positive pregnant women are further tested for HBeAg and HBV viral load to assess the risk of vertical transmission. However, HBIG administration is not a routine practice for children born to HBV infected mothers in Russia. Recommended by WHO tenofovir treatment to prevent the vertical transmission is not yet applied in Russia. So, the only intervention for children born to HBV infected mothers is four doses immunization at months 0, 1, 2 and 12 compared to standard vaccination schedule for children born to HBsAg negative mothers (0, 1 and 6 months).  [Russian National Vaccination Schedule, available at official site of the Russian Ministry of Healthcare at https://minzdrav.gov.ru/opendata/7707778246-natskalendarprofilakprivivok2015/visual]. We added the respective paragraph to Discussion section of the revised manuscript (page 11, lines 342-352).
